# Specific binding of Hsp27 and phosphorylated Tau mitigates abnormal Tau aggregation-induced pathology

Shengnan Zhang[1†], Yi Zhu[2†], Jinxia Lu[3†], Zhenying Liu[1,4†], Amanda G Lobato[2,5], Wen Zeng[1,4], Jiaqi Liu[2,6], Jiali Qiang[1], Shuyi Zeng[3], Yaoyang Zhang[1], Cong Liu[1], Jun Liu[7], Zhuohao He[1], R Grace Zhai[2]*, Dan Li[3,8]*

[1]Interdisciplinary Research Center on Biology and Chemistry, Shanghai Institute of Organic Chemistry, Chinese Academy of Sciences, Shanghai, China; [2]Department of Molecular and Cellular Pharmacology, University of Miami Miller School of Medicine, Miami, United States; [3]Bio-X Institutes, Key Laboratory for the Genetics of Developmental and Neuropsychiatric Disorders (Ministry of Education), Shanghai Jiao Tong University, Shanghai, China; [4]University of the Chinese Academy of Sciences, Beijing, China; [5]Graduate Program in Human Genetics and Genomics, University of Miami Miller School of Medicine, Miami, United States; [6]Graduate Program in Molecular and Cellular Pharmacology, University of Miami Miller School of Medicine, Miami, United States; [7]Department of Neurology and Institute of Neurology, Ruijin Hospital, Shanghai Jiao Tong University School of Medicine, Shanghai, China; [8]Zhangjiang Institute for Advanced Study, Shanghai Jiao Tong University, Shanghai, China

*For correspondence:
gzhai@med.miami.edu (RGZ);
lidan2017@sjtu.edu.cn (DL)

[†]These authors contributed equally to this work

Competing interest: The authors declare that no competing interests exist.

**Abstract** Amyloid aggregation of phosphorylated Tau (pTau) into neurofibrillary tangles is closely associated with Alzheimer's disease (AD). Several molecular chaperones have been reported to bind Tau and impede its pathological aggregation. Recent findings of elevated levels of Hsp27 in the brains of patients with AD suggested its important role in pTau pathology. However, the molecular mechanism of Hsp27 in pTau aggregation remains poorly understood. Here, we show that Hsp27 partially co-localizes with pTau tangles in the brains of patients with AD. Notably, phosphorylation of Tau by microtubule affinity regulating kinase 2 (MARK2), dramatically enhances the binding affinity of Hsp27 to Tau. Moreover, Hsp27 efficiently prevents pTau fibrillation in vitro and mitigates neuropathology of pTau aggregation in a *Drosophila* tauopathy model. Further mechanistic study reveals that Hsp27 employs its N-terminal domain to directly interact with multiple phosphorylation sites of pTau for specific binding. Our work provides the structural basis for the specific recognition of Hsp27 to pathogenic pTau, and highlights the important role of Hsp27 in preventing abnormal aggregation and pathology of pTau in AD.

## Editor's evaluation

Phosphorylated Tau (pTau) aggregation into neurofibrillary tangles is closely associated with Alzheimer's disease (AD), and this process is mediated by several molecular chaperones. The authors dissect the important molecular mechanism of the interplay between Hsp27 and pTau, which is relevant to pathology, using a combination of approaches including a *Drosophila* tauopathy model, and biophysical and computational methods. This work provides fundamental molecular insights into the important role of Hsp27 in preventing pTau pathology in AD.

## Introduction

Pathological aggregation and propagation of microtubule-associated protein Tau is closely associated with Alzheimer's disease (AD), frontotemporal dementia (FTD), and other tauopathies (*Avila, 2006*; *Hanger et al., 2009*; *Hanger et al., 1998*; *Hanger et al., 1991*; *Spillantini and Goedert, 2013*). Tau is hyper-phosphorylated and forms amyloid fibrils in neurofibrillary tangles (NFT) in patients' brains which is the pathological hallmark of AD. Tau is abundant in the axons of neurons where it stabilizes microtubules (MT; *Drechsel et al., 1992*; *Drubin and Kirschner, 1986*). Under diseased conditions, abnormal phosphorylation of Tau in the regions surrounding the microtubule-binding domain by kinases such as microtubule affinity regulating kinase (MARK) and glycogen synthase kinase 3 (GSK3) leads to dissociation of Tau from MT and subsequent pathological aggregation (*Ando et al., 2016*; *Biernat et al., 1993*; *Drewes, 2004*; *Drewes et al., 1997*; *Hanger et al., 2009*; *Martin et al., 2013*). Moreover, the amyloid fibrils formed by phosphorylated Tau (pTau) have been found to be more potent in mediating the propagation and spread of Tau pathology than those formed by unphosphorylated Tau (*Dujardin et al., 2020*; *Hu et al., 2016*; *Rosenqvist et al., 2018*). Pathological pTau fibrils have been extracted from AD brains, and Hsp27 has been found to co-precipitate with them (*Shimura et al., 2004*). This suggests that Hsp27 may play an important role in pTau pathology.

Hsp27 is a member of the small heat shock protein (sHsp) family which participates in the cellular chaperone network to maintain protein homeostasis by preventing protein abnormal aggregation (*Haslbeck et al., 2005*). It is ubiquitously expressed and plays a protective role in a variety of cellular processes (*Jakob et al., 1993*). Emerging evidence suggests the importance of Hsp27 in AD and other tauopathies. For instance, the expression level of Hsp27 is significantly elevated in affected brain tissues of patients with AD (*Björkdahl et al., 2008*; *Renkawek et al., 1994*), as well as patients with other tauopathies including progressive supranuclear palsy (PSP) and corticobasal degeneration (CBD) (*Schwarz et al., 2010*). Moreover, Hsp27 can effectively inhibit the amyloid aggregation of unphosphorylated Tau in vitro (*Baughman et al., 2018*; *Baughman et al., 2020*; *Freilich et al., 2018*), prevent cellular toxicity of Tau in SH-SY5Y cells (*Choi et al., 2015*), and rescue tauopathies in Tau transgenic mice (*Abisambra et al., 2010*). However, the molecular mechanism of the interplay between Hsp27 and pTau, which exhibits more pathological relevance than that between Hsp27 and unphosphorylated Tau, remains poorly understood.

In this study, we first examined the postmortem brain tissue of patients with AD and detected the co-localization of Hsp27 and pTau aggregates. We next took advantage of a *Drosophila* tauopathy model and uncovered the neuroprotective effects of Hsp27 against pTau-induced synaptopathy in vivo. To dissect the molecular mechanism, we characterized the interaction between Hsp27 and MARK2-mediated hyper-phosphorylated pTau, and the consequent amyloid aggregation in vitro. Employing multiple biophysical and computational approaches, we revealed that Hsp27 recognizes multiple phosphorylation sites of pTau to prevent pTau amyloid aggregation. This work provides molecular insights into the important role of Hsp27 in preventing pTau pathology in AD.

## Results

### Co-localization of Hsp27 and Tau pathology in human brains with AD

We first asked whether Hsp27 is associated with pathological pTau aggregates in AD brains. We obtained human frontal cortex tissue slices from two cases of AD patients and two cases of age-matched healthy controls (*Table 1*). Immunofluorescence staining using antibodies anti-pTau[Ser262] and anti-Hsp27 detected the presence of pTau in two AD cases, but absent in two healthy controls (*Figure 1*). Notably, Hsp27 was detected at a high level in AD brain slices, but at a much lower or undetectable level in healthy control brain slices (*Figure 1*), consistent with previous findings of elevated Hsp27 levels in AD brains (*Björkdahl et al., 2008*; *Renkawek et al., 1994*). More importantly, Hsp27 was found to co-localize with the pTau aggregates in both AD cases (*Figure 1*), suggesting the pathological relevance between Hsp27 and pTau pathology in AD. Note that not all pTau aggregates contain Hsp27 and vice versa, especially in case 1, which may reflect the heterogeneity of the protein aggregates.

**Table 1.** Summary of the AD patients and age-matched healthy controls.

| Cases | Gender | Age | Brain region | Thal Phase (0–3, Aβ plaques) | Braak stage (0–6, Tau) | CERAD (0–3, neuritic plaque) |
|---|---|---|---|---|---|---|
| AD case 1 | F | 76 | Frontal cortex | 3 | 3 | 3 |
| AD case 2 | F | 74 | Frontal cortex | 3 | 3 | 3 |
| Normal case 1 | M | 77 | Frontal cortex | 0 | 0 | 0 |
| Normal case 2 | F | 75 | Frontal cortex | 0 | 0 | 0 |

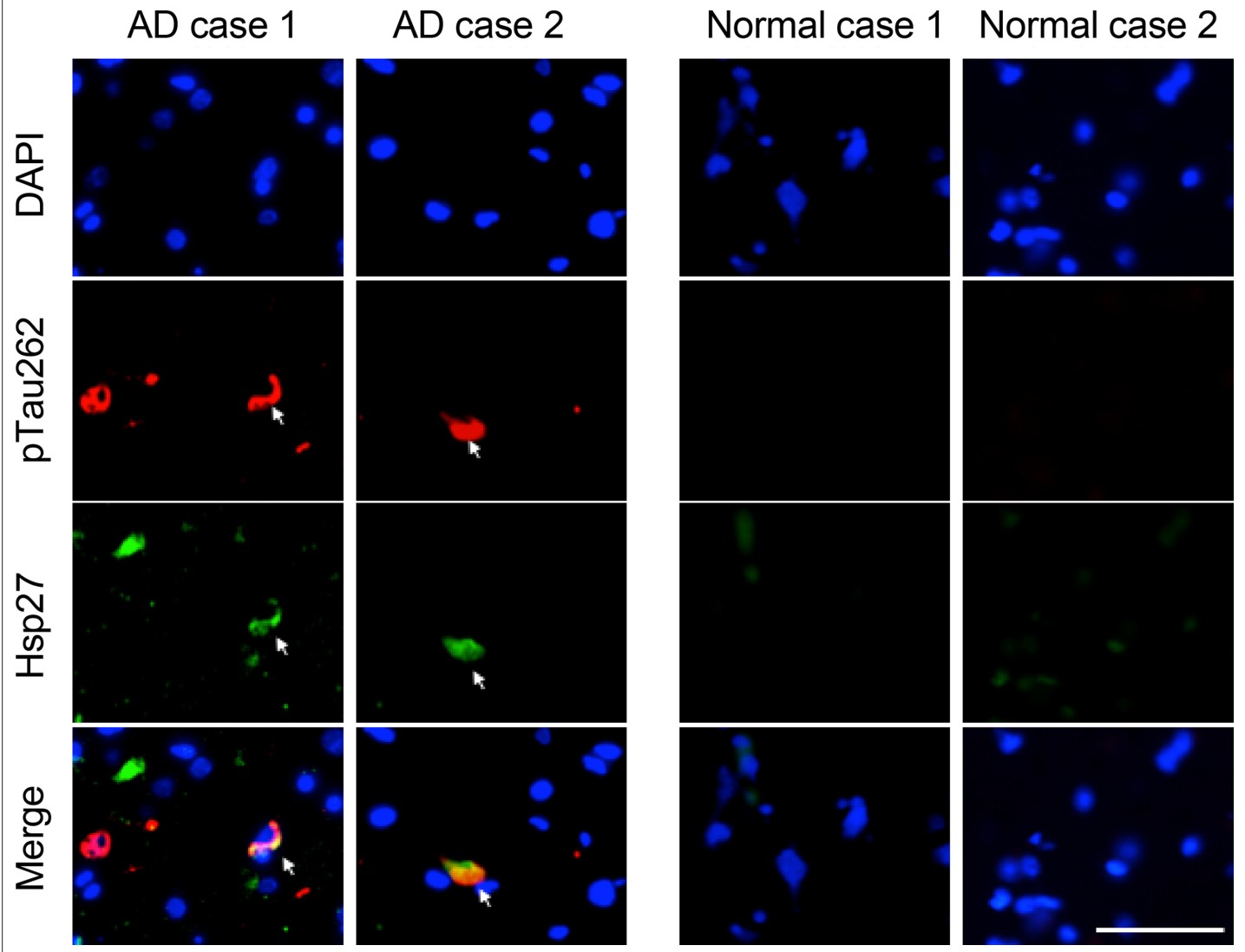

**Figure 1.** Hsp27 partially co-localizes with pTau aggregates in the brains of AD patients. Representative images of immunofluorescence staining using anti-hyper-phosphorylated Tau at Ser262 and anti-Hsp27 antibodies on the brain slices from two AD and two age-matched normal cases. Green, Hsp27; red, pTau[S262]; blue, DAPI; Scale bar, 50 μm.

## Hsp27 protects pTau-induced synaptopathy in *Drosophila*

To investigate the influence of Hsp27 on the abnormal aggregation of pTau and its neuropathology in vivo, we used a *Drosophila* tauopathy model where human pathogenic mutant Tau (Tau[R406W]), associated with FTD with parkinsonism linked to chromosome 17 (FTDP-17), is expressed in the nervous system by a pan-neuronal driver *elav[C155]-GAL4*. Our previous work has demonstrated that the expression of human Tau[R406W] in the *Drosophila* nervous system leads to age-dependent neurodegeneration recapitulating some of the salient features of tauopathy in FTDP-17 (*Ali et al., 2012*; *Ma et al., 2020*). To quantitatively assess the pTau species in the brain, we carried out western blot analysis using a total Tau antibody 5A6 (*Johnson et al., 1997*), a pTau[Ser262] specific antibody (*Iijima et al., 2010*), and a hyper-phosphorylated Tau antibody AT8 that recognizes hyper-phosphorylation at Ser202 and Thr205 sites (*Goedert et al., 1995*). Hsp27 overexpression significantly reduced hyper-phosphorylated Tau levels 2 and 10 days after eclosion (DAE) (*Figure 2A and B*, *Figure 2—source data 1*; *Figure 2—source data 2*).

We next examined the morphology of the fly brain and the accumulation of hyper-phosphorylated Tau by immunofluorescence staining. We used the AT8 antibody to label hyper-phosphorylated Tau because phosphorylation at Ser202 and Thr205 occurs after the phosphorylation of S262 and

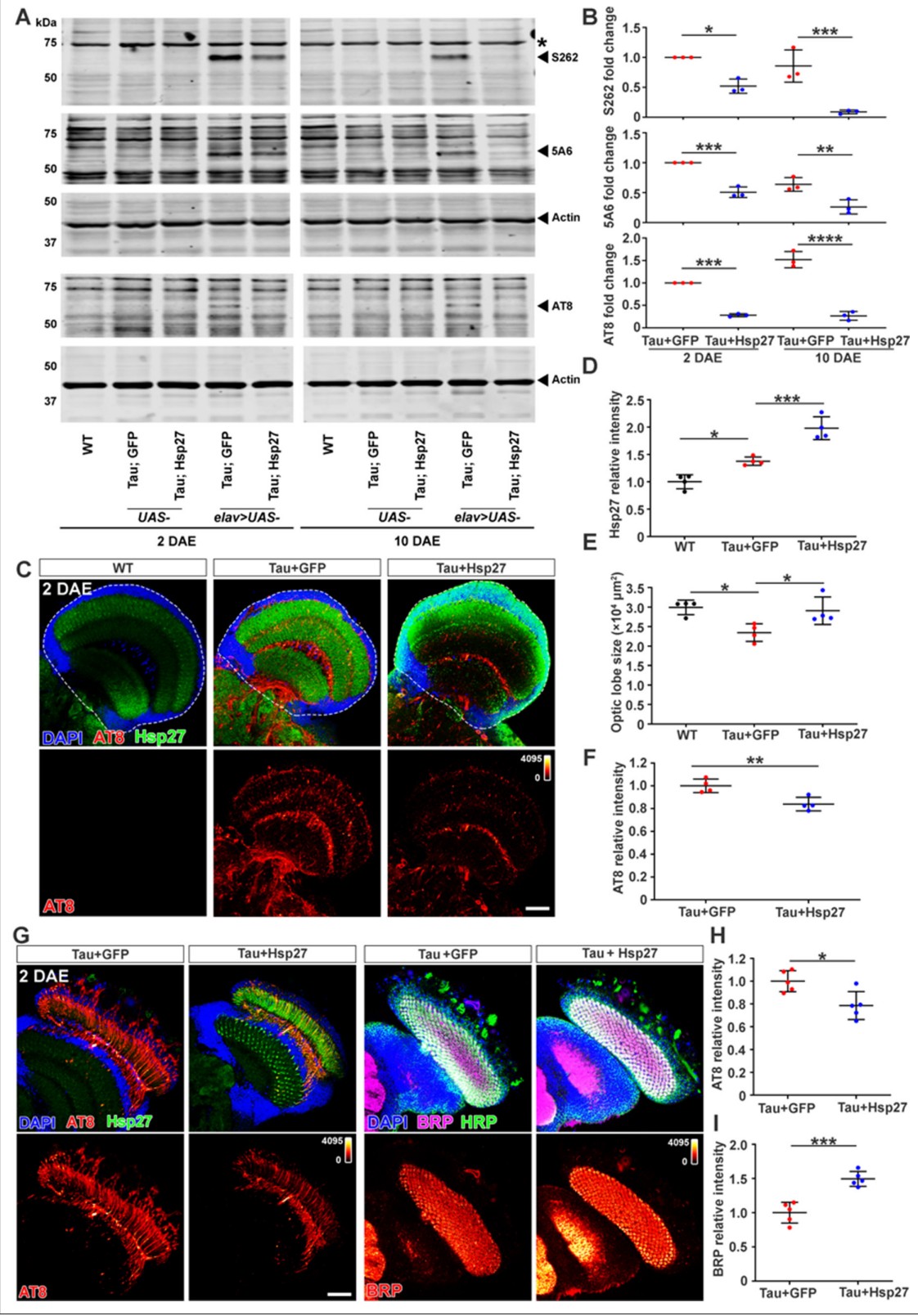

**Figure 2.** Hsp27 reduces pTau level and protects against pTau-induced synaptopathy in *Drosophila*. (**A**) Brain lysates of 2 and 10 days after eclosion (DAE) wild-type (WT) flies (lanes 1 and 6), flies expressing human Tau with GFP (lanes 4 and 9), or human Tau with Hsp27 (lanes 5 and 10) in the nervous system were probed with antibodies for disease-associated phospho-tau epitopes S262, Ser202/Thr205 (AT8), and total Tau (5A6). Actin was probed as a loading control. Brain lysates of flies carrying only UAS elements were loaded for control (lanes 2, 3, 7, and 8). (**B**) Quantification of protein fold

*Figure 2 continued on next page*

*Figure 2 continued*

changes in (**A**). The levels of Tau species were normalized to actin. Fold changes were normalized to the Tau +GFP group at 2 DAE. n=3. (**C**) Brains of WT flies or flies expressing Tau +GFP or Tau +Hsp27 in the nervous system at 2 DAE were probed for AT8 (heatmap) and Hsp27 (green), and stained with DAPI (blue). Scale bar, 30 µm. (**D–F**) Quantification of the Hsp27 intensity (**D**, data normalized to WT), brain optic lobe size (**E**), and AT8 intensity (**F**, data normalized to the Tau +GFP group). n=4. (**G**) Brain of flies expressing Tau +GFP or Tau +Hsp27 in photoreceptors were probed for AT8, Hsp27, bruchpilot (BRP), and horseradish peroxidase (HRP). Scale bar, 30 µm. (**H, I**) Quantification of AT8 intensity (**H**) and BRP intensity (**I**). Data were normalized to the Tau +GFP group. n=5. Statistical analyses were performed using one-way ANOVA with Bonferroni's post hoc test (**B, D, E**) or independent samples t-test (**F, H, I**). All data are presented as mean ± SD. *p<0.05, **p<0.01, ***p<0.001, ****p<0.0001.

The online version of this article includes the following source data for figure 2:

**Source data 1.** The full blots for *Figure 2A*.

**Source data 2.** Quantification of protein fold changes.

is associated with advanced pathology (*Wesseling et al., 2020*). Consistent with previous findings, brains with neuronal expression of Tau[R406W] exhibited an accumulation of filamentous pTau and a reduction of brain neuropil size indicative of neurodegeneration (*Figure 2C–F*). Interestingly, we found that neuronal expression of Tau[R406W] led to a significant upregulation of endogenous Hsp27, which was highly enriched in the synaptic areas (*Figure 2C and D*), suggesting that Hsp27 might be a part of the neuronal stress response network that is upregulated upon proteotoxic stress. Importantly, overexpression of Hsp27 restored the size of brain neuropil and suppressed the accumulation of filamentous pTau (*Figure 2C–F*), suggesting that Hsp27 protects against mutant Tau[R406W] induced neurodegeneration.

Synaptic loss is a hallmark of human tauopathy (*McGowan et al., 2006*). To evaluate the synaptic integrity, we took advantage of the *Drosophila* visual system with highly organized paralleled photoreceptor structures (*Fischbach and Dittrich, 1989*). We have previously shown that Tau[R406W] expression resulted in synaptic aggregation of hyper-phosphorylated Tau as well as a significant reduction of Bruchpilot (Brp), an active zone associated-cytoskeletal matrix protein, at lamina cartridge, suggesting a severe loss of the active zone structures in the presynaptic terminals (*Ma et al., 2020*). As shown in *Figure 2G–I*, we expressed Tau[R406W] with either GFP (control) or Hsp27 in the photoreceptors using *GMR-GAL4* and found that overexpression of Hsp27 led to a remarkable reduction of synaptic pTau and enhanced BRP localization, indicating a restoration of synaptic integrity.

Taken together, we showed that Hsp27 protects against synaptic dysfunction in a *Drosophila* tauopathy model by reducing pTau aggregation and ameliorating pTau-induced synaptic degeneration.

## Hsp27 specifically binds pTau and prevents its amyloid aggregation

We next sought to examine the binding of Hsp27 with pTau and its effect on pTau amyloid aggregation in vitro. We purified the longest Tau isoform – Tau40 that contains both the projection region and four microtubule-binding repeat domains (R1–R4), as well as K19, a truncated Tau construct that contains R1, R3, and R4 (*Figure 3A*). Then phosphorylated Tau, including pTau40 and pK19, were prepared by phosphorylating purified Tau40 and K19 using kinase MARK2 which was previously identified to be important in mediating pTau pathology in AD (*Ando et al., 2016*; *Drewes, 2004*; *Gu et al., 2013*). Two dimensional (2D) $^1$H-$^{15}$N HSQC spectra enable us to monitor each of the MARK2 phosphorylation sites in both pTau40 and pK19. Consistent with previous report (*Schwalbe et al., 2013*), eight MARK2 phosphorylation sites including pS262, pS293, pS305, pS324, pS352, pS356, pS413, and pS416 were identified on the HSQC spectrum of pTau40 (*Figure 3—figure supplement 1A*). Four phosphorylation sites including pS262, pS324, pS352, and pS356 were identified on the HSQC spectrum of pK19 (*Figure 3—figure supplement 1B*). Most of these phosphorylation sites are within the fibril forming core region of Tau, and two of them including pS262 and pS356 have been widely reported to be essential for the detachment of pTau from MT and cause increased toxicity in animal models (*Hanger et al., 1998*; *Hanger et al., 2009*; *Ali et al., 2012*).

We used BioLayer Interferometry (BLI) assay to measure the binding affinity of Hsp27 with pTau and unphosphorylated Tau, respectively. pTau and Tau were immobilized on biosensor tips, and the association and dissociation curves were measured in the presence and absence of various concentrations of Hsp27 to calculate the equilibrium dissociation constant ($K_D$) between Hsp27 and pTau/Tau. Notably, the $K_D$ value of Hsp27/pTau40 is ~1.58 µM (*Figure 3B*; *Figure 3—source data 1*). The $K_D$ value for Hsp27/pK19 is ~1.98 µM (*Figure 3C*; *Figure 3—source data 1*) which is similar to that of

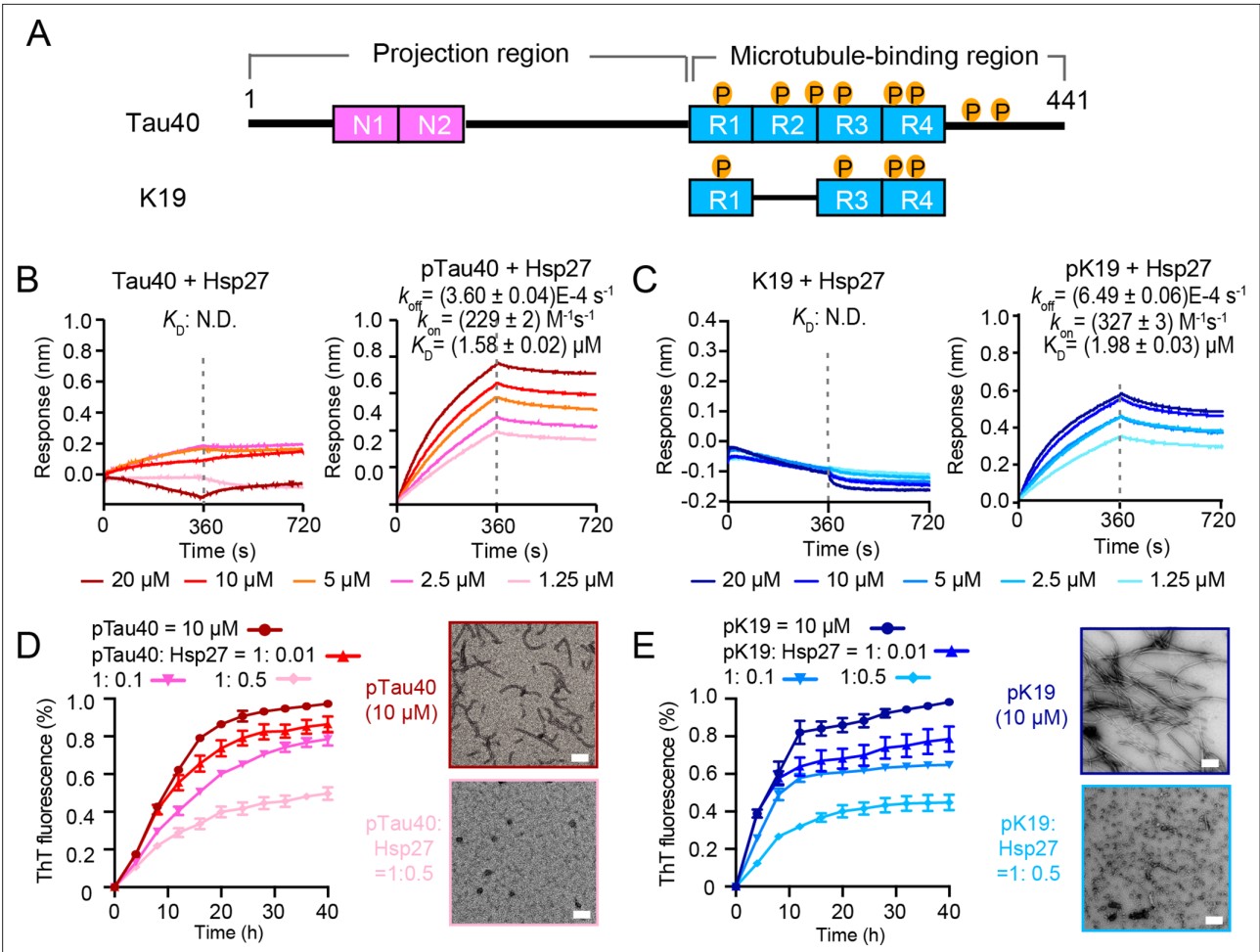

**Figure 3.** Hsp27 specifically binds MARK2-phosphorylated Tau and prevents its amyloid aggregation. (**A**) Domain schematic of the longest isoform of Tau (Tau40) and a truncated construct K19. Different isoforms of Tau are characterized by 0, 1, or 2 N-terminal insertions (**N1 and N2**), and three (**R1–R3–R4**) or four (**R1-4**) microtubule-binding repeats. The MARK2-phosphorylated sites are indicated. (**B**) Binding affinity of Tau40/pTau40 with Hsp27 determined by BLI. The association and dissociation profiles of Tau40/pTau40 to Hsp27 were divided by a vertical dash line. Tau40/pTau40 was fixed to the sensor, and the 5 concentrations of Hsp27 used are indicated. N.D., not detectable. The determined equilibrium constant ($K_D$), and the association ($k_{on}$) and dissociation ($k_{off}$) rates are labelled. (**C**) Binding affinity of K19/pK19 with Hsp27 determined by BLI. The determined equilibrium constant ($K_D$), and the association ($k_{on}$) and dissociation ($k_{off}$) rates are labelled. (**D&E**) Inhibition of Hsp27 on the amyloid aggregation of 10 µM pTau40 (**D**)/pK19 (**E**) revealed by ThT fluorescence kinetic assay (left) and TEM microscopy (right). A gradient concentration of Hsp27 was applied as indicated. The ThT data showed correspond to mean ± SEM, with n=3 technical replicates. The top and bottom panel on the right are the TEM images of 10 µM pTau40 (**D**)/pK19 (**E**), and 10 µM pTau40 (**D**)/pK19 (**E**) with Hsp27 at a molar ratio of 1:0.5 taken at the end point of the ThT kinetic assay on the left, respectively. Scale bar in TEM images, 50 nm.

The online version of this article includes the following source data and figure supplement(s) for figure 3:

**Source data 1.** The association and dissociation response of Tau40/pTau40 and K19/pK19 with Hsp27 (*Figure 3B and C*).

**Figure supplement 1.** 2D $^1$H-$^{15}$N HSQC spectra of pTau40 (**A**) and pK19 (**B**) collected at 298 K on a Bruker Avance 900 MHz spectrometer.

Hsp27/pTau40. In addition, both the association ($k_{on}$) and dissociation ($k_{off}$) rates of Hsp27 to pK19 are at the same order of magnitude of Hsp27 to pTau40. These results suggest that pK19 serves as the key region for binding of Hsp27 to pTau40. In sharp comparison, the binding of both Hsp27/Tau40 and Hsp27/K19 were much weaker which cannot be detectable under our experimental condition (*Figure 3B and C*; *Figure 3—source data 1*). These results demonstrate that hyper-phosphorylation significantly enhances the binding of Hsp27 to both pTau40 and pK19.

We further assessed the influence of Hsp27 on the amyloid aggregation of pTau40 and pK19 by thioflavin T (ThT) fluorescence assay and negative-staining electron microscopy (EM). The results showed that Hsp27 effectively inhibited the amyloid aggregation of both pTau40 and pK19 in a

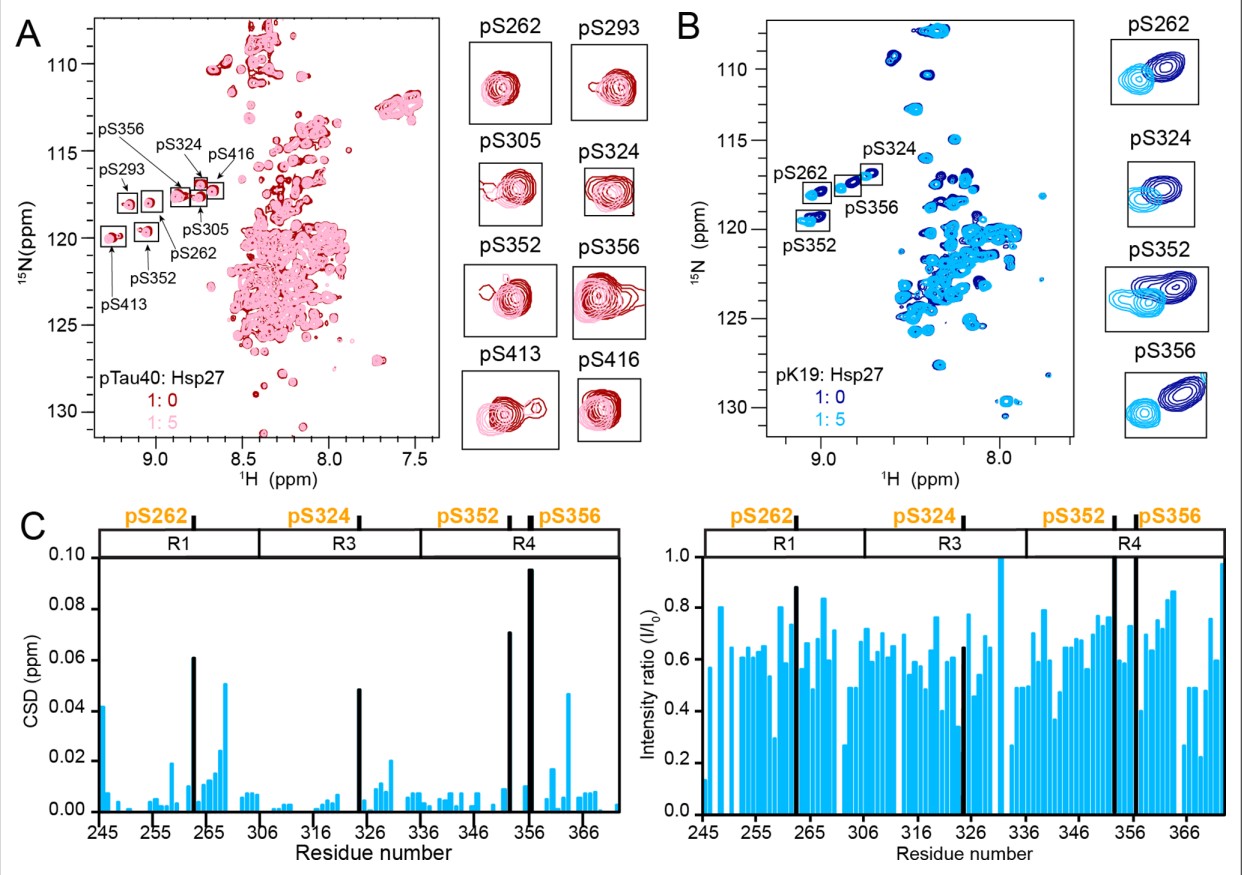

**Figure 4.** The phosphorylation sites of pTau strongly interact with Hsp27. (A&B) Overlay of the 2D ${}^{1}$H-${}^{15}$N HSQC spectra of 50 μM pTau40 (**A**) and 50 μM pK19 (**B**) in the absence and presence of 250 μM Hsp27. Signals of pSer residues are enlarged and labeled on the right. (**C**) Residue-specific CSDs (left) and intensity changes (right) of pK19 titrated by Hsp27 from (**B**). The domain organization of pK19 is indicated on top and the data of pSer residues are labeled.

The online version of this article includes the following source data and figure supplement(s) for figure 4:

**Source data 1.** Residue-specific chemical shift changes and intensity changes of pK19 titrated by Hsp27.

**Figure supplement 1.** Hsp27 interacts very weakly with unphosphorylated Tau.

**Figure supplement 1—source data 1.** Residue-specific chemical shift changes and intensity changes of K19 titrated by Hsp27.

dose-dependent manner (*Figure 3D and E*; *Figure 3—source data 1*). Together, our data show that Hsp27 can specifically bind to hyper-phosphorylated Tau including pTau40 and pK19, and efficiently prevent their amyloid aggregation.

## Hsp27 directly binds to multiple phosphorylation sites of pTau

To investigate the structural basis of the interaction between Hsp27 and pTau, we performed solution NMR spectroscopy, and titrated Hsp27 to ${}^{15}$N-labeled pTau40 and ${}^{15}$N-labeled pK19, respectively. Strikingly, the 2D ${}^{1}$H-${}^{15}$N HSQC spectra showed obvious chemical shift changes (CSDs) of all the eight phosphorylated Ser (pSer) residues of pTau40 and four pSer residues of pK19 upon titration of Hsp27 (*Figure 4A and B*). The highly overlapped signals in the pTau40 HSQC spectrum exclude us to obtain a full assignment of pTau40. Alternatively, we accomplished the backbone assignment of pK19, which enables us to map the binding interface of pK19 with Hsp27. As shown in *Figure 3C*, the regions containing the four pSer residues of pK19 harbor prominent CSDs. Among them, the two pSer residues in R4 showed the largest CSDs above 0.06 ppm (*Figure 4C*; *Figure 4—source data 1*). In addition, titration of Hsp27 to pK19 induced dramatically intensity drops of most residues of pK19 (*Figure 4C*; *Figure 4—source data 1*), implying the direct binding of Hsp27 to pK19 which increases the molecular weight upon pK19-Hsp27 complex formation, and thus increases line width of the cross

peaks. By contrast, addition of the same amount of Hsp27 to unphosphorylated Tau including Tau40 and K19 only results in slightly CSD and intensity perturbations (*Figure 4—figure supplement 1*; *Figure 4—figure supplement 1—source data 1*). Together, our data suggest that phosphorylation of Tau promotes the interaction between Hsp27 and pTau by forming the specific interaction between Hsp27 and the phosphorylated residues of Tau.

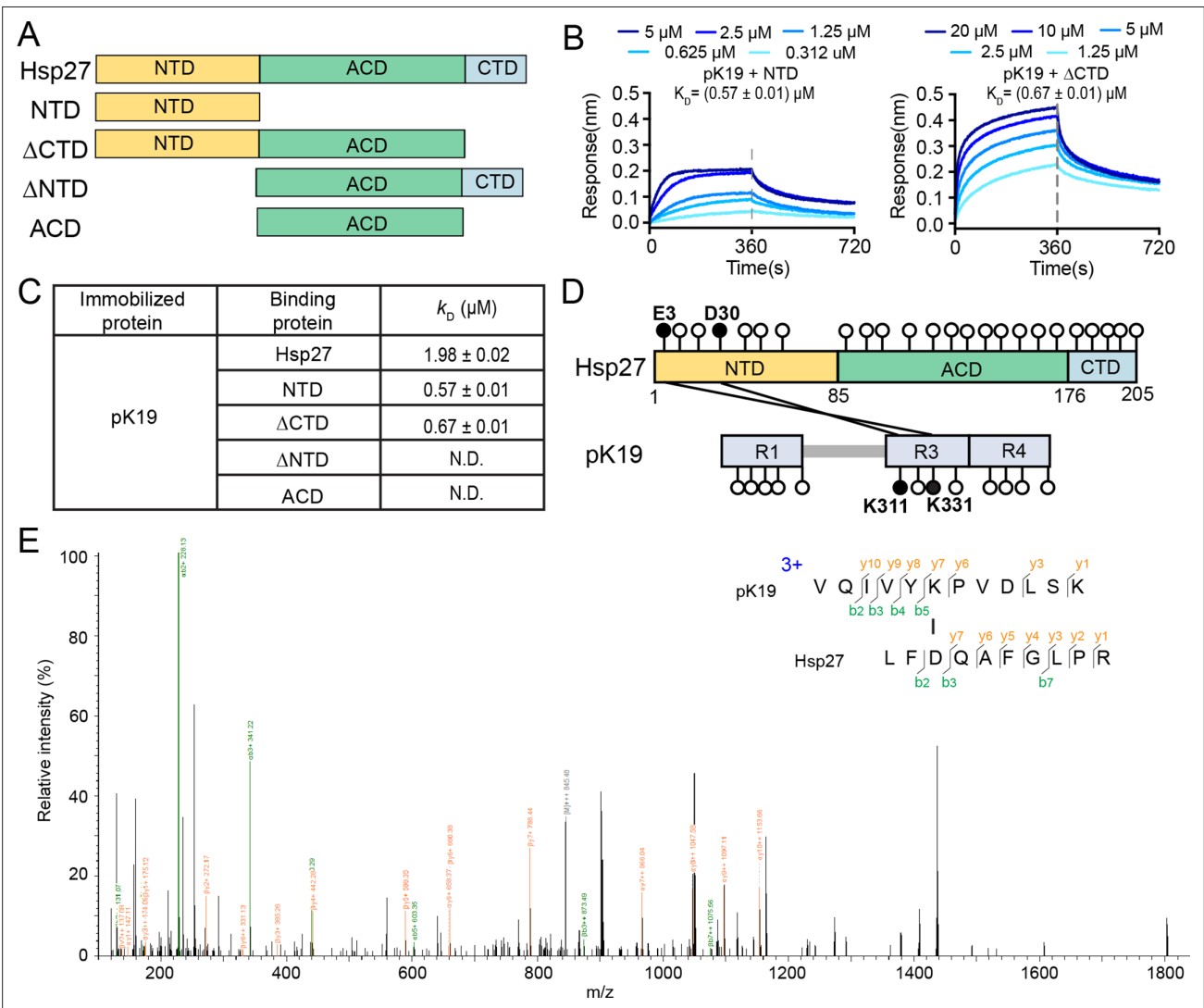

**Figure 5.** N-terminal domain of Hsp27 is essential in binding with pK19. (**A**) Domain organization of Hsp27 and the four truncations. (**B**) Binding affinity of pK19 with NTD and ΔCTD of Hsp27 determined by BLI. The association and dissociation profiles were divided by a vertical dash line. pK19 was fixed to the sensor, and the 5 concentrations of Hsp27 truncations used are indicated. (**C**) Summary of the binding affinity of pK19 with Hsp27 wild type and truncations. (**D**) Schematic profile of the cross-linked results of Hsp27 to pK19 using cross-linkers EDC and NHS. All GLU (**E**) and ASP (**D**) residues in Hsp27, and all LYS (**K**) residues in pK19 are indicated by circles, respectively. The two identified cross-linked segments are indicated by two black lines and the corresponding residues are highlighted in black circles and labeled. (**E**) A representative MS/MS spectrum of trypsin proteinase-generated peptide. The m/z of fragment ions were matched to their theoretical values generated by in silico fragmentation.

The online version of this article includes the following source data and figure supplement(s) for figure 5:

**Source data 1.** The association and dissociation response of pK19 with NTD and ΔCTD of Hsp27.

**Figure supplement 1.** Binding affinity of pK19 with ΔNTD and ACD of Hsp27 determined by BLI.

**Figure supplement 1—source data 1.** The association and dissociation response of pK19 with ΔNTD and ACD of Hsp27.

**Table 2.** Cross-linked peptides between pK19 and Hsp27.
The cross-linked residues are bold and underlined in the sequences.

| | Peptide sequence (Hsp27-pK19) | Score 1 | Score 2 | Score 3 |
|---|---|---|---|---|
| 1 | LF**D**QAFGLPR -VQIVY**K**PVDLSK | 2.56246E-11 | 3.17803E-09 | 1.21049E-06 |
| 2 | GSEFENLYFQGMT**E**R-CGSLGNIHH**K**PGGGQVEVK | 5.05514E-09 | 1.37169E-06 | 9.56423E-07 |

## N-terminal domain (NTD) of Hsp27 mediates the binding of Hsp27 to pTau

Hsp27 composes of a central α-crystallin domain (ACD) flanked by a flexible NTD and a flexible C-terminal domain (CTD). To further dissect the role of different domains of Hsp27 in binding pK19, four different truncations of Hsp27 were prepared (*Figure 5A*). Their binding affinities to pK19 were determined by BLI, respectively. Notably, the isolated NTD binds to pK19 with a $K_D$ value of ~0.57 μM which is similar to the binding affinity of the full-length Hsp27 to pK19 (*Figure 5B*; *Figure 5—source data 1*). ΔCTD which contains both NTD and ACD display a similar binding affinity to pK19 (0.67±0.01 μM; *Figure 5B*; *Figure 5—source data 1*). In contrast, both ACD and ΔNTD which do not contain NTD show no detectable binding to pK19 at the same condition (*Figure 5C* and *Figure 5—figure supplement 1*; *Figure 5—figure supplement 1—source data 1*). Together, these data suggest that Hsp27 NTD plays an essential role in mediating the binding of Hsp27 to pK19.

Next, we performed cross-linking mass spectrometry (CL-MS) to map the interacting regions of Hsp27 and pK19. pK19 was used since it contains less amino acids but keeps the key region of pTau40 for binding Hsp27 to simplify the data analysis. NTD of Hsp27 contains 0 lysine residue. Whereas, negatively charged residues including glutamine acid (E) and aspartic acid (D) evenly distribute within the three different domains of Hsp27 (*Figure 5D*). Thus, we performed a two-step coupling procedure using cross-linkers 1-ethyl-3-(3-dimethylaminopropyl) carbodiimide (EDC) and sulfo-N-hydroxysulfosuccinimide (sulfo-NHS) to cross-link Hsp27 and pK19. EDC causes direct conjugation of carboxylates (-COOH) to primary amines (-NH$_2$) of target molecules, and sulfo-NHS is introduced for stabilization. We identified two pairs of cross-linked segments between Hsp27 and pK19 with a confidence score of <10$^{-6}$ by mass spectrometry, including E3$_{Hsp27}$-K331$_{pK19}$ and D30$_{Hsp27}$-K311$_{pK19}$ (*Figure 5D and E* and *Table 2*). Notably, both cross-linked segments are formed by residues located at R3 of pK19 and NTD of Hsp27 (*Figure 5D*), which is consistent with our NMR titration that residues at R3 show slightly stronger intensity drop upon addition of Hsp27, and the domain truncation mapping results. Together, our data demonstrate that NTD is essential to mediate pK19 binding.

## Different domains of Hsp27 bind distinct regions of pK19 to prevent its fibrillation

To further pinpoint the binding regions of pK19 by NTD and other domains of Hsp27, we titrated full-length and the four truncations of Hsp27 to $^{15}$N-labeled pK19 and collected a series of HSQC spectra. Titration of both NTD and ΔCTD induce significant CSDs mainly located on the four pSer sites of pK19 (*Figure 6A and B*; *Figure 6—source data 1*), which recapitulates that titrated by full-length Hsp27. In contrast, titration of ACD or ΔNTD caused much smaller CSDs on the four pSer sites of pK19. Moreover, addition of NTD and ΔCTD induces overall intensity drops of pK19 signals (~I/I$_0$<0.8), while titration of ACD and ΔNTD only cause slightly intensity drops. These results demonstrate that Hsp27 mainly uses its NTD to directly interact with the pSer residues of pK19 for pTau binding, which is consistent with the BLI results.

Notably, we found that despite that titration of ACD does not induce CSDs of the pSer sites, it causes relative small but detectable CSDs (>0.01 ppm) and intensity drops (I/I$_0$<0.7) of the highly amyloidogenic region - $^{306}$VQIVYK$^{311}$ of pK19, especially residues V309, Y310, and K311 (*Figure 6A and B*; *Figure 6—source data 1*). This demonstrates that ACD is capable of directly binding to pK19 via its VQIVYK region. Of note, the cross peaks of V306, Q307, and I308 severely overlapped with a few residues of pK19 which excludes us to obtain the accurate CSD and intensity changes upon titration. Together, our results demonstrate that different domains of Hsp27 display distinct binding

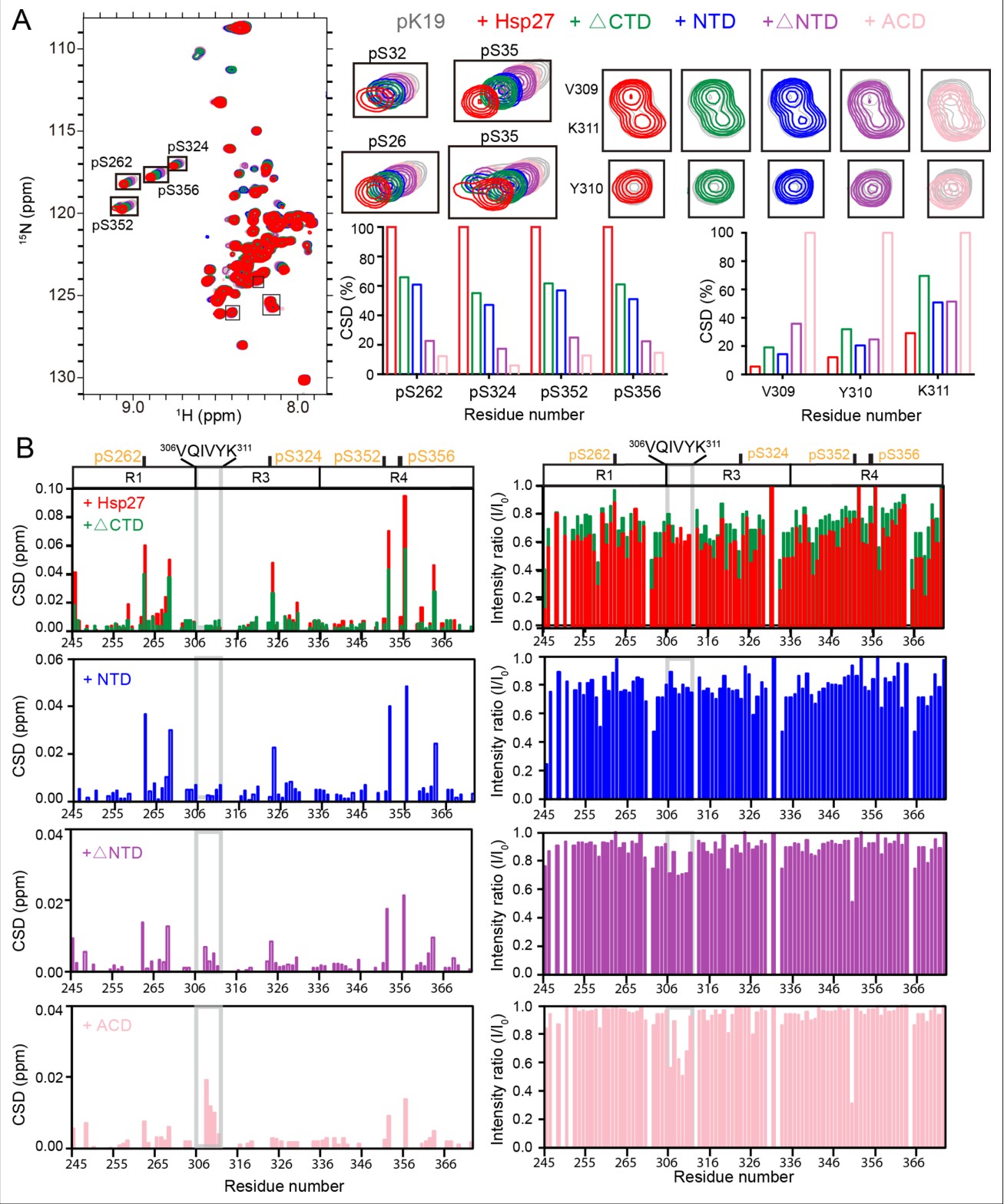

**Figure 6.** Interaction between pK19 and different domains of Hsp27. (**A**) Overlay of the 2D $^1$H-$^{15}$N HSQC spectra of 50 μM pK19 in the absence and presence of 100 μM Hsp27 variants as indicated. Signals of the four pSer residues and residues V309, Y310, and K311 are labeled and enlarged on the right. Relative CSDs of the four pSer residues and residues V309, Y310, and K311 of pK19 titrated by Hsp27 variants are shown on the right. (**B**) Residue-specific CSDs (left) and intensity changes (right) of pK19 titrated by Hsp27 variants from (**A**). Domain organization of pK19 is indicated on top and the pSer residues are indicated. The amyloidogenic $^{306}$VQIVYK$^{311}$ region of pK19 was labeled and highlighted by light gray lines.

*Figure 6 continued on next page*

*Figure 6 continued*

The online version of this article includes the following source data for figure 6:

**Source data 1.** Residue-specific chemical shift changes and intensity changes of pK19 titrated by Hsp27 variants.

preferences to different regions of pK19. NTD exhibits a high binding affinity to the multiple pSer sites of pK19, while ACD weakly binds the amyloidogenic VQIVYK segment of pK19.

Finally, we examined the role of different domains of Hsp27 in preventing amyloid aggregation of pK19 and pTau40. As shown in *Figure 7A and B* (*Figure 7—source data 1*), NTD-containing Hsp27 variants including full-length Hsp27, ΔCTD, and NTD exhibits potent inhibitory activity in preventing both pTau40 and pK19 fibrillation. This suggests that strong binding of NTD to the phosphorylated sites of pTau is important in preventing pTau fibrillation. Interestingly, ACD and ΔNTD display weakened but not negligible inhibitory activity of pTau, especially for pK19. This suggests that binding of ACD to the VQIVYK segment of pTau may result in moderate inhibition of pTau fibrillation. Together, our data show that the NTD is predominately employed to bind pTau to prevent its fibrillation, while ACD also contributes chaperone activity to a less degree by binding to the highly aggregation-prone VQIVYK region of pTau.

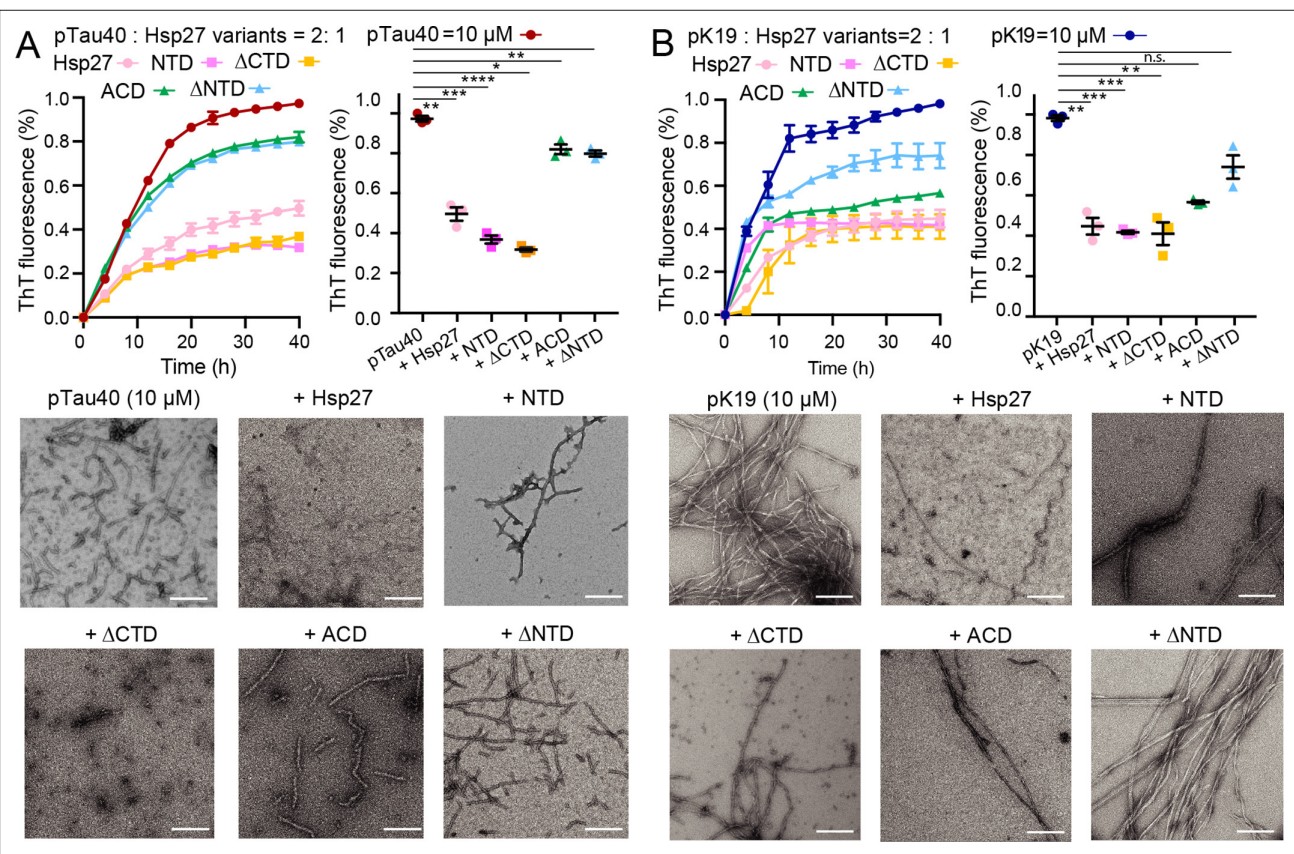

**Figure 7.** Different domains of Hsp27 play distinctive roles in preventing pTau amyloid aggregation. (**A&B**) Inhibition of Hsp27 variants on the amyloid aggregation of 10 μM pTau40 (**A**)/pK19 (**B**) revealed by ThT fluorescence kinetic assay (top) and TEM microscopy (bottom). The ThT data showed correspond to mean ± s.d., with n=3 technical replicates. Comparison of the inhibitory effect of Hsp27 variants on pTau aggregation at 40 hr time point is shown on the right. p Values based on two-sided Student's t-test. *p<0.05, **p<0.01, ***p<0.001, ****p<0.0001, n.s., not significant. Scale bar in TEM images, 200 nm.

The online version of this article includes the following source data for figure 7:

**Source data 1.** ThT fluorescence profiles of pTau40 (*Figure 7A*) and pK19 (*Figure 7B*) in the absence or presence of Hsp27 variants at the indicated concentrations.

## Discussion

Molecular chaperones play essential roles in maintaining protein homeostasis and preventing protein misfolding and aggregation. Accumulating evidence showed that many molecular chaperones including Hsp70, DnaJA2, Hsp90, Hsp27, and NMNAT can directly interact with Tau and are involved in different steps of Tau biogenesis, aggregation, and clearance (*Baughman et al., 2018*; *Freilich et al., 2018*; *Ma et al., 2020*; *Mok et al., 2018*; *Petrucelli et al., 2004*). Since Hsp27 was identified to co-precipitate with AD brain-derived pTau aggregates and is elevated in AD brains (*Björkdahl et al., 2008*; *Renkawek et al., 1994*; *Shimura et al., 2004*), Hsp27 may play an important role in interacting with pTau in AD. Indeed, we showed elevated Hsp27 levels and partial co-localization of Hsp27 with pTau aggregates in AD brains. More importantly, we found that MARK2-mediated phosphorylation of Tau dramatically increases the binding of pTau to Hsp27, strengthening the notion that Hsp27 binds specifically to disease-related pTau. This specific binding enables Hsp27 to be a potent chaperone in inhibiting amyloid aggregation of MARK2-mediated phosphorylated Tau in vitro.

Over 58 different phosphorylation sites including MARK2-phosphorylation sites were identified on Tau40 in AD brains (*Hanger et al., 1998*; *Hanger et al., 2007*; *Hasegawa et al., 1992*; *Morishima-Kawashima et al., 1995*; *Wesseling et al., 2020*; *Xia et al., 2021*). Phosphorylation at different sites is believed to be involved in different stages of pTau pathology in AD (*Hanger et al., 2009*; *Wesseling et al., 2020*; *Xia et al., 2021*). A previous study examined the inhibitory activity of different chaperons to unmodified Tau and Tau carrying single-point mutations at different phosphorylation sites including S356E, T153E, T231E, S396E, and S404E (*Mok et al., 2018*). Among them, only S356 belongs to MARK2-phosphorylation sites (*Figure 4A*). Strikingly, Hsp27 exhibits potent inhibitory activity on S356E Tau but not the other Tau phosphomimetics (*Mok et al., 2018*). This suggests the binding specificity of Hsp27 on different phosphorylation sites of Tau. It will be important to further explore whether Hsp27 recognizes the other disease-related phosphorylation sites in addition to those modified by MARK2. Moreover, other molecular chaperones including Hsp90 and NMNAT were also found to mediate pTau cellular clearance and prevent pTau aggregation both in vitro and in cells (*Dickey et al., 2007*; *Ma et al., 2020*). The important questions that remained to be addressed include: (1) whether different chaperones preferentially recognize Tau with different phosphorylation patterns; (2) how different chaperones coordinate to prevent disease-related pTau aggregation and pathology.

Hsp27 contains three domains including NTD, ACD, and CTD. We showed in this study that Hsp27 uses its NTD to strongly bind to the multiple phosphorylation sites of pTau. Meanwhile, ACD of Hsp27 transiently and weakly interacts with the highly amyloidogenic region - $^{306}$VQIVYK$^{311}$ of pTau. Therefore, different domains of Hsp27 are responsible for binding to distinct regions of pTau. Notably, previous studies showed that, as for the unphosphorylated Tau, ACD exhibits the strongest binding ability by predominantly binding the $^{306}$VQIVYK$^{311}$ region of Tau via its β4 and β8 region (*Baughman et al., 2018*; *Baughman et al., 2020*; *Freilich et al., 2018*). While NTD displays much weaker binding to unmodified Tau (*Baughman et al., 2020*; *Freilich et al., 2018*). Therefore, phosphorylation of Tau shifts the paradigm of Hsp27 binding pattern of Tau, which results in significantly enhanced overall binding of Hsp27 to pTau. As the phosphorylation pattern of pTau is dynamically changed upon the progression of AD (*Hanger et al., 2009*; *Wesseling et al., 2020*; *Xia et al., 2021*), how phosphorylation at different sites influences the interplay between pTau and its binding patterns including Hsp27 and different molecular chaperones will be interesting to investigate.

This study demonstrates that NTD of Hsp27 is essential for binding to the phosphorylation sites of pTau. Notably, Hsp27 was found to be phosphorylated at multiple sites on NTD under stress conditions, which leads to dissociation of Hsp27 from high-molecular-weight oligomers to smaller ones with higher chaperone activities (*Alderson et al., 2019*; *Jovcevski et al., 2017*; *Jovcevski et al., 2015*; *Lambert et al., 1999*; *Rogalla et al., 1999*). Our previous work showed that NTD phosphorylation provides an additional layer for regulating Hsp27 activity in controlling the phase separation and amyloid fibrillation of ALS-associated FUS (*Liu et al., 2020*). Whether Hsp27 NTD is phosphorylated under the disease conditions of AD, FTD, and other tauopathies, and how Hsp27 NTD phosphorylation influences the binding and chaperone activity of Hsp27 to pTau are both worthy of being further explored to fully appreciate the complicated interplay between pTau and Hsp27 in disease. Nevertheless, our study demonstrates the important role of Hsp27 in chaperoning pathological pTau from abnormal aggregation, and implies that activating Hsp27 or elevating its level might be a potential strategy for AD treatment.

## Materials and methods
### Human brain samples and immunofluorescence
Human brain samples were originally from the tissue bank of the Center for Neurodegenerative Disease Research (CNDR) at the University of Pennsylvania. 10% neutral buffered formalin (NBF)-fixed, paraffin-embedded human frontal cortex tissues from two AD patients and two age-matched normal controls were sectioned into 6 µm slices, and immunostained as described before (*He et al., 2018*). Briefly, after deparaffinization and rehydration, the brain slices were incubated with anti-human Hsp27 (1:200, Cat. # ab2790) and anti-pTau$^{Ser262}$ (1:200, Cat. # ab131354) antibodies overnight at 4 °C followed by a 2 hr incubation of Alexa Fluor-conjugated secondary antibodies (Thermo Fisher Scientific), and then mounted with DAPI-containing Fluoromount-G (SouthernBiotech, USA).

### *Drosophila* stocks and genetics
The following fly strains were used: *UAS-Hsp27* (generated in this study); *UAS-Tau$^{R406W}$* obtained from Dr. Mel Feany (*Wittmann et al., 2001*); *yw*, *UAS-GFP*, *elav-GAL4*, *GMR-GAL4* obtained from Bloomington *Drosophila* Stock Center. Flies were reared on cornnmeal-molasses-yeast medium at 22°C, 65% humidity, with 12 hr light/12 hr dark cycles.

### Western blot
For protein extraction in flies, 10 heads of each group were homogenized in radioimmunoprecipitation assay (RIPA) buffer (Sigma-Aldrich, R0278). Samples were mixed with Laemmli sample buffer containing 2% SDS, 10% glycerol, 62.5 mM Tris-HCl (pH 6.8), 0.001% bromophenol blue, and 5% β-mercaptoethanol, and heated at 95 °C for 10 min. Proteins were separated by SDS-polyacrylamide gel electrophoresis and transferred to a nitrocellulose membrane. After blocking at room temperature for 1.5 hr, the membrane was incubated with primary antibody at 4 °C overnight, followed by secondary antibody for 1.5 hr at room temperature. Imaging was performed on an Odyssey Infrared Imaging system (LI-COR Biosciences) and analyzed using Image Studio (ver 4.0). Primary antibody dilutions were used as follows: anti-pTau$^{Ser202/Thr205}$ (AT8, 1:500; ThermoScientific, Rockford, IL, USA), anti-pTau$^{Ser262}$ (1:1000; ThermoScientific), anti-β-actin (1:5000, Sigma-Aldrich, St Louis, MO, USA), anti-total-Tau (5A6, 1:250, Developmental Studies Hybridoma Bank, Iowa City, IA, USA).

### Fly brain dissection, immunostaining, and confocal microscopy
Fly brains were dissected in phosphate-buffered saline (PBS, pH 7.4) and fixed with 4% formaldehyde at room temperature for 10 min. After 10 min washing in PBTx (PBS containing 0.4% v/v Triton X-100) three times, brains were incubated with primary antibodies diluted in PBTx with 5% normal goat serum at 4 °C overnight. Fly brains were incubated with conjugated secondary antibodies at room temperature for 1 hr, followed by 4′,6- diamidino-2-phenylindole (DAPI, 1:300, Invitrogen, Carlsbad, CA, USA) staining for 15 min. Brains were mounted on slides with VECTASHIELD Antifade Mounting Medium (Vector Laboratories Inc, Burlingame, CA, USA). After that, brains were imaged using an Olympus IX81 confocal microscope coupled with a 60×oil immersion objective lens with a scan speed of 8.0 µs per pixel and spatial resolution of 1024×1024 pixels. Images were processed using FluoView 10-ASW (Olympus) and Adobe Photoshop CS6 and quantified using Fiji/Image J (ver 1.52). Primary antibody dilutions were used as follows: anti-pTau$^{Ser202/Thr205}$ (AT8, 1:250), anti-BRP (nc82, 1:250, Developmental Studies Hybridoma Bank), anti-Hsp27 (1:250, Abcam, Cambridge, MA, USA). The experimenter was blinded to the genotype during confocal scans. The sample size was not predetermined by statistical calculations, but the number of flies and independent biological replicates were large enough to achieve sufficient statistical power, according to our previous publications (*Ma et al., 2020*; *Zhu et al., 2019*). All the data were included in the final analysis.

### Plasmid construction for in vitro study
All plasmids used in this study are available upon request. Genes of full-length human Hsp27 (UniProt accession number P04792), and the truncations of Hsp27 including NTD (residues of 1–84), ΔCTD (residues of 1–176), ΔNTD (residues of 85–205), and ACD (residues of 85–175) were inserted into a pET-28a vector with an N-terminal His$_6$-tag and a following tobacco etch virus (TEV) protease cleavage site. Sequencing of all constructs was verified by GENEWIZ company (Suzhou, China).

## Protein expression and purification

All proteins were expressed in *Escherichia coli* BL21(DE3) cells, grown to an $OD_{600}$ of 0.8 and induced with 0.3 mM IPTG overnight at 16 °C. Hsp27 and its variants were purified with HisTrapFF column (GE Healthcare) with the Tris buffer (50 mM Tris-HCl, 100 mM NaCl, a gradient of 0~500 mM imidazole, pH 8.0). The N-terminal $His_6$-tag was removed using TEV protease in the buffer of 50 mM Tris-HCl, 100 mM NaCl, pH 8.0. The cleaved proteins were immediately loaded onto the size-exclusion chromatography column Superdex 75 26/60 (GE Healthcare) with a PBS buffer of 50 mM sodium phosphate, 50 mM NaCl at pH 7.0.

Human Tau40 and K19 were over-expressed and purified as previously described (*Barghorn et al., 2005*). Briefly, Tau/K19 was purified by a HighTrap HP SP (5 ml) column (GE Healthcare), and followed by a Superdex 75 gel filtration column (GE Healthcare).

For $^{15}N$-labeled proteins, protein expression was the same as that for unlabeled proteins except that the cells were grown in M9 minimal medium with $^{15}NH_4Cl$ (1 g $l^{-1}$). Purification of $^{15}N$-labeled proteins was the same as that of the unlabeled proteins.

The purity of proteins was assessed by SDS-PAGE. Protein concentration was determined by BCA assay (Thermo Fisher).

## In vitro phosphorylation of Tau40/K19

Phosphorylation of Tau40/K19 by MARK2 kinase was carried out following a method described previously (*Ma et al., 2020*). Briefly, Tau40/K19 was incubated with cat MARK2-T208E at a molar ratio of 10:1 in a buffer of 50 mM Hepes, pH 8.0, 150 mM KCl, 10 mM $MgCl_2$, 5 mM ethylene glycol tetraacetic acid (EGTA), 1 mM PMSF, 1 mM dithio-threitol (DTT), 2 mM ATP (Sigma), and protease inhibitor cocktail (Roche) at 30 °C overnight. pTau40/pK19 was further purified by HPLC (Agilent) to remove kinase, and lyophilized. The sites of phosphorylation were quantified using 2D $^1H$-$^{15}N$ HSQC spectrum according to previous publications (*Schwalbe et al., 2013*).

## NMR spectroscopy

All NMR experiments were performed at 298 K in the NMR buffer of 50 mM sodium phosphate, 50 mM NaCl, and 10% (v/v) $D_2O$ at pH 7.0. 3D HNCA and HNCACB experiments were collected on an Agilent 600 MHz spectrometer for backbone assignment of K19, while 3D HNCA, HNCOCA, CBCACONH, and HNCACB experiments were collected for backbone assignment of pK19. NMR titrations were performed on a Bruker 900 M or Agilent 800 MHz spectrometer. Each sample (500 μl) was made of 50 μM $^{15}N$-pTau40/pK19/Tau40/K19, in the absence or presence of Hsp27 and its variants at the indicated concentrations. Intensity changes were calculated by $I/I_0$. And chemical shift deviations (CSDs, $\Delta\delta$) were calculated using equation,

$$\Delta\delta = \sqrt{\left(\Delta\delta 1H\right)^2 + 0.0289 \left(\Delta\delta 15N\right)^2}$$

Where $\Delta\delta 1H$ and $\Delta\delta 15N$ are the chemical shift differences of amide proton and amide nitrogen between free and bound state of the protein, respectively. All NMR spectra were processed using NMRPipe (*Delaglio et al., 1995*), and analyzed using Sparky (*Lee et al., 2015*) and NMRView (*Johnson, 2004*).

## BioLayer interferometry (BLI) assay

The binding kinetics of the Tau proteins to Hsp27 WT and variants were measured by BLI on an ForteBio Octet RED96 system (Pall ForteBio LLC). Experiments were performed at room temperature using the assay buffer of 50 mM sodium phosphate, 50 mM NaCl, pH 7.0. Tau proteins including pTau40, Tau40, pK19, and K19 were firstly biotinylated by incubating 0.5–1 mg/ml proteins with biotin at a molar ratio of protein: biotin of 2:3 at 4 °C for 30 min, then the excess biotins were removed by desalting column (Zeba Spin Desalting Columns, Thermo). Then biotinylated Tau proteins were immobilized onto streptavidin biosensors (ForteBio) individually, and incubated with varying concentrations of Hsp27 WT and variants as indicated in the figure. The kinetic experiments were performed following the protocol previously (*Zhang et al., 2021*), in which an auto-inhibition step was used to eliminate the non-specifically binding of Hsp27 and its variants to biosensors. The resulting curves were corrected using the blank reference and analyzed by the ForteBio Data Analysis software 9.0.

## ThT fluorescence assay

ThT kinetics of pTau/pK19 in the absence and presence of Hsp27 and its variants were recorded using a Varioskan Flash Spectral Scanning Multimode Reader (Thermo Fisher Scientific) with sealed 384-microwell plates (Greiner Bio-One). The assay buffer is 30 mM PBS, 2 mM MgCl2, 1 mM DTT, 0.05% NaN$_3$, pH 7.4. 0.5% (v/v) of fibril seeds (the seeds were prepared by sonicating fibrils for 15 s) were added to promote the fibril formation of pTau/pK19. ThT fluorescence with a final ThT concentration of 30 μM in each sample was measured in triplicates with shaking at 600 *rpm* at 37 °C with excitation at 440 nm and emission at 485 nm.

## Transmission electron microscopy (TEM)

Five μl of samples were applied to fresh glow-discharged 300-mesh copper carbon grids and stained with 3% v/v uranyl acetate. Specimens were examined by using Tecnai G2 Spirit TEM operated at an accelerating voltage of 120 kV. Images were recorded using a 4K × 4K charge-coupled device camera (BM-Eagle, FEI Tecnai).

## Cross-linking mass spectrometry analysis

Cross-linking experiments were modified from the protocol previously (*Gong et al., 2015*). 0.4 mg EDC (final concentration of ~2 mM) and 1.1 mg Sulfo-NHS (final concentration of ~5 mM) were added to 1 ml of Hsp27 solution and reacted for 15 mins at room temperature. Then 1.4 μl 2-mercaptoethanol (final concentration of 20 mM) was added to quench the activity of EDC. Then pK19 was added to the reaction system at an equal molar ratio with Hsp27, and the proteins were allowed to react for 2 hr at room temperature. Then Tris was added to the reaction system with a final concentration of 50 mM to quench the reaction. Cross-linking products were analyzed by SDS-PAGE to assess the cross-linking efficiency. Before MS analysis, proteins were precipitated with acetone; the pellet was resuspended in 8 M urea, 100 mM Tris (pH 8.5) and digested with trypsin at room temperature overnight. The resulting peptides were analyzed by online nanoflow liquid chromatography tandem mass spectrometry (LC−MS/MS). And the mass spectrometry data were analyzed by pLink (*Yang et al., 2012*).

## Acknowledgements

We thank staff members of the National Facility for Protein Science in Shanghai, Zhangjiang Laboratory, China for providing technical support and assistance in NMR and BLI data collection. This work was supported by the National Natural Science Foundation (NSF) of China (Grant No. 82188101, 32170683, 31872716 and 32171236), the Science and Technology Commission of Shanghai Municipality (STCSM) (Grant No. 20XD1425000 and 2019SHZDZX02), CAS project for Young Scientists in Basic research (Grant No. YSBR-009), the Shanghai Pilot Program for Basic Research – Chinese Academy of Science, Shanghai Branch (Grant No. CYJ-SHFY-2022–005), the Joint Funds of the National Natural Science Foundation of China (Grant No. U1932204). This research was also supported in part by the Florida Department of Health (FDOH) grant 21A21 (to RGZ) and National Institute of Health (NIH) R01NS109640 (to RGZ).

# Additional information

## Funding

| Funder | Grant reference number | Author |
| --- | --- | --- |
| National Natural Science Foundation of China | 82188101 | Cong Liu |
| National Natural Science Foundation of China | 32170683 | Dan Li |
| National Natural Science Foundation of China | 31872716 | Cong Liu<br>Dan Li |
| National Natural Science Foundation of China | 32171236 | Cong Liu |

| Funder | Grant reference number | Author |
| --- | --- | --- |
| Science and Technology Commission of Shanghai Municipality | 20XD1425000 | Cong Liu |
| Science and Technology Commission of Shanghai Municipality | 2019SHZDZX02 | Cong Liu |
| Chinese Academy of Sciences | Project for Young Scientists in Basic Research YSBR-009 | Shengnan Zhang |
| Chinese Academy of Sciences | Shanghai Pilot Program for Basic Research CYJ-SHFY-2022-005YSBR-009 | Cong Liu |
| National Natural Science Foundation of China | Joint Funds U1932204 | Shengnan Zhang |
| Florida Department of Health | 21A21 | R Grace Zhai |
| National Institutes of Health | R01NS109640 | R Grace Zhai |

The funders had no role in study design, data collection and interpretation, or the decision to submit the work for publication.

## Author contributions

Shengnan Zhang, Conceptualization, Data curation, Formal analysis, Methodology, Writing – original draft, Writing – review and editing; Yi Zhu, Data curation, Formal analysis, Methodology, Writing – original draft, Writing – review and editing; Jinxia Lu, Zhenying Liu, Conceptualization, Data curation, Formal analysis, Methodology, Writing – original draft; Amanda G Lobato, Jiaqi Liu, Jiali Qiang, Shuyi Zeng, Data curation, Formal analysis; Wen Zeng, Data curation, Formal analysis, Writing – review and editing; Yaoyang Zhang, Supervision, Methodology; Cong Liu, Dan Li, Conceptualization, Supervision, Funding acquisition, Investigation, Project administration, Writing – review and editing; Jun Liu, Formal analysis, Writing – review and editing; Zhuohao He, Resources, Supervision, Investigation, Writing – review and editing; R Grace Zhai, Supervision, Funding acquisition, Investigation, Project administration, Writing – review and editing

## Author ORCIDs

Shengnan Zhang ![ORCID] http://orcid.org/0000-0003-1051-7196
Yi Zhu ![ORCID] http://orcid.org/0000-0002-1778-8880
Amanda G Lobato ![ORCID] http://orcid.org/0000-0002-4218-7817
Cong Liu ![ORCID] http://orcid.org/0000-0003-3425-6672
R Grace Zhai ![ORCID] http://orcid.org/0000-0002-7599-1430
Dan Li ![ORCID] http://orcid.org/0000-0002-1609-1539

## Ethics

Human subjects: Post-mortem human brain tissue sections were gifted from the Brain bank of the Center for Neurodegenerative Disease Research (CNDR) at the University of Pennsylvania with material transfer agreement and necessary sample information. The tissues were collected at CNDR following a standard procedure, which can be referred to Toledo et al., Alzheimer's & Dementia, 10 (2014) 477-484. The use of those samples was approved by the Ethical committee at Interdisciplinary Research Center on Biology and Chemistry, Chinese Academy of Sciences.

## Decision letter and Author response

Decision letter https://doi.org/10.7554/eLife.79898.sa1
Author response https://doi.org/10.7554/eLife.79898.sa2

# Additional files

## Supplementary files
- MDAR checklist

## Data availability

All data generated or analysed during this study are included in the manuscript and supporting file; source data files have been provided for Figures 2–7.

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
