## [Editor Report]

Phosphorylated Tau (pTau) aggregation into neurofibrillary tangles is closely associated with Alzheimer's disease (AD), and this process is mediated by several molecular chaperones. The authors dissect the important molecular mechanism of the interplay between Hsp27 and pTau, which is relevant to pathology, using a combination of approaches including a *Drosophila* tauopathy model, and biophysical and computational methods. This work provides fundamental molecular insights into the important role of Hsp27 in preventing pTau pathology in AD.

---

## [Decision Letter]

**Decision letter after peer review:**

Thank you for submitting your article "Specific binding of Hsp27 and phosphorylated Tau mitigates abnormal Tau aggregation-induced pathology" for consideration by *eLife*. Your article has been reviewed by 2 peer reviewers, including Keqiang Ye as the Reviewing Editor and Reviewer #1, and the evaluation has been overseen by Jeannie Chin as the Senior Editor.

Essential revisions:

The manuscript is well organized and the data basically support the main conclusion. It is recommended that alternative complementary tools that confirm the observations in the manuscript would substantially improve the quality. Complementary assays including both immunofluorescent or immunohistochemical staining and immunoblotting with different antibodies etc., would be extremely valuable to validate the findings.

There are several concerns that need to be addressed (see below), which would require further experiments and analyses, especially in the *Drosophila* tauopathy model part in Figure 2.

1) In Figure 1 normal case #2, the nuclei staining seems odd.

2) The Figure 2 legend doesn't match its contents. Labels in figure panels are not readable.

3) The binding of Hsp27 with pTau40 and pK19 was investigated. Could you purify phosphorylated full-length Tau with three repeats: R1, R3, and R4 and look at the potential difference of binding with Hsp27 compared to pTau40?

4) Is there a way to quantify the difference between pTau40 + Hsp27 vs pK19 + Hsp27 in the BLI assay?

5) Please describe the TEM microscope images in Figure 3D and E? Are we looking at tau fibrils on the top panels and tau monomers in the lower panels?

6) The mapping of different domains of Hsp27 to different regions of pK19 is fascinating and promising. Could you please justify why pK19 was chosen, and not pTau40?

---

## [Author Response]

Essential revisions:The manuscript is well organized and the data basically support the main conclusion. It is recommended that alternative complementary tools that confirm the observations in the manuscript would substantially improve the quality. Complementary assays including both immunofluorescent or immunohistochemical staining and immunoblotting with different antibodies etc., would be extremely valuable to validate the findings.There are several concerns that need to be addressed (see below), which would require further experiments and analyses, especially in the *Drosophila* tauopathy model part in Figure 2.

We sincerely thank the editors and reviewers for the positive remarks, and the insightful and helpful suggestions which are important to improve this manuscript. We very seriously took the suggestions and addressed them with additional data or discussion. We are confident that the revised manuscript is much strengthened.

1) In Figure 1 normal case #2, the nuclei staining seems odd.

Thanks for pointing this out. The normal case #2 sample was stored in formalin for a prolonged time. Thus, it is possible that the formalin could be oxidized into formic acid which may destroy the nuclear acid structure and lead to low binding for DAPI. To address this issue, we used a new normal case which shows normal staining pattern. The image of the new normal case (normal case 2) was placed in Figure 1. Accordingly, the information of the normal case 2 was updated in Table 1 in the revised manuscript.

Representative images of immunofluorescence staining using anti-hyper-phosphorylated Tau at Ser262 and anti-Hsp27 on the brain slices from two AD and two age-matched normal cases. Green, Hsp27; red, pTau^S262^; blue, DAPI; Scale bar, 50 μm.

2) The Figure 2 legend doesn't match its contents. Labels in figure panels are not readable.

We sincerely apologize for this. And we are sorry that we did not supply a high-quality Figure 2 in the previous manuscript which might confuse the reviewers. In the revised manuscript, we replaced Figure 2 with a high-quality image (Figure 2).

3) The binding of Hsp27 with pTau40 and pK19 was investigated. Could you purify phosphorylated full-length Tau with three repeats: R1, R3, and R4 and look at the potential difference of binding with Hsp27 compared to pTau40?

Thanks for raising this very important point. To fully address this concern, we cloned and purified Tau23 which is the full-length Tau containing three repeats: R1, R3, and R4. We further used MARK2 kinase to prepare phosphorylated Tau23 (pTau23). Then, BLI assay was conducted to determine the binding affinity of pTau23 to Hsp27 by following the same protocol as that for pTau40 and Hsp27 binding measurement. As shown in Author response image 1, the *K*_D_ value of Hsp27 to pTau23 is ~ 0.68 µM, which is similar to that of Hsp27 to pTau40 (~1.58 µM). This result further confirms that Hsp27 binds to phosphorylated Tau containing either four repeats or three repeats with similar high binding affinities.

**Author response image 1. sa2fig1:** Binding affinity of pTau23 with Hsp27 determined by BLI. The association and dissociation profiles of pTau23 to Hsp27 were divided by a vertical dash line. pTau23 was fixed to the sensor, and the 5 concentrations of Hsp27 used are indicated.

4) Is there a way to quantify the difference between pTau40 + Hsp27 vs pK19 + Hsp27 in the BLI assay?

We thank the reviewing editor for this question. To address this, we reanalyse the BLI data, the resulting equilibrium dissociation constant (*K*_D_) between Hsp27 to pTau40 and pK19 are ~1.58 and 1.98 µM (Figure 3b and C), respectively. In addition, we calculated the association (k_on_) and dissociation (k_off_) rates determined by the BLI assay. As shown in Figure 3B and C, the association rate for Hsp27 to pTau40 and pK19 are ~229 and 323 M^-1^s^-1^, while the dissociation rate for Hsp27 to pTau40 and pK19 are 3.60*10^-4^ and 6.38*10^-4^ s^-1^, respectively. Although there is no significant difference of the association and dissociation rates between Hsp27 to pTau40 and pK19, it seemed that pK19 (327 M^-1^s^-1^) associates slightly faster than pTau40 (228 M^-1^s^-1^) to Hsp27, and pK19 (6.49*10^-4^ s^-1^) also dissociates slightly faster than pTau40 (3.60*10^-4^ s^-1^) from Hsp27. Thus, the final equilibrium dissociation constant (*K*_D_ = k_off_/ k_on_) between Hsp27 to pTau40 and pK19 were at the same order of magnitude, and no significant difference between them was observed. Accordingly, we added the k_off_ and k_on_ values in Figure 3B and C in the revised manuscript, and added the description in the revised Results, section “Hsp27 specifically binds pTau and prevents its amyloid aggregation”, paragraph 2, as following:

“The *K*_D_ value for Hsp27/pK19 is ~ 1.98 µM (Figure 3C) which is similar to that of Hsp27/pTau40 (~ 1.58 µM). In addition, both the association (k_on_) and dissociation (k_off_) rates of Hsp27 to pK19 are at the same order of magnitude of that Hsp27 to pTau40. These results suggest that pK19 serves as the key region for binding of Hsp27 to pTau40.”

5) Please describe the TEM microscope images in Figure 3D and E? Are we looking at tau fibrils on the top panels and tau monomers in the lower panels?

Thanks for pointing this out. We are sorry that we did not make it clear in the previous manuscript. The top panel on the right of Figure 3D is the TEM image of the ThT sample of pTau40 (10 μM) taken at the end point of the ThT kinetic assay on the left, in which obvious pTau40 fibrils were observed. The bottom panel on the right of Figure 3D is the TEM image of the ThT sample of pTau40 (10 μM) with Hsp27 at a molar ratio of 1:0.2 (pTau40 vs Hsp27) taken at the end point of the ThT kinetic assay on the left, in which only very few fibrils were observed. These TEM results are consistent with the ThT results that addition of Hsp27 efficiently inhibits the amyloid aggregation of pTau40.

Similar, the top panel on the right of Figure 3E is the TEM image of the ThT sample of pK19 (10 μM) taken at the end point of the ThT kinetic assay on the left, in which obvious pK19 fibrils were observed. The bottom panel on the right of Figure 3E is the TEM image of the ThT sample of pK19 (10 μM) with Hsp27 at a molar ratio of 1:0.2 (pK19 vs Hsp27) taken at the end point of the ThT kinetic assay on the left, in which only very few fibrils were observed. These TEM results are also consistent with the ThT results that addition of Hsp27 efficiently inhibits the amyloid aggregation of pK19.

We revised the figure legend of Figure 3D and E in the revised manuscript as following:

“(DandE) Inhibition of Hsp27 on the amyloid aggregation of 10 μM pTau40 (D)/pK19 (E) revealed by ThT fluorescence kinetic assay (left) and TEM microscopy (right). A gradient concentration of Hsp27 used in the ThT assay was applied as indicated. The ThT data showed correspond to mean ± SEM, with n = 3. The top and bottom panel on the right are the TEM images of 10 μM pTau40 (D)/pK19 (E), and 10 μM pTau40 (D)/pK19 (E) with Hsp27 at a molar ratio of 1:0.5 taken at the end point of the ThT kinetic assay on the left, respectively. Scale bar in TEM images, 50 nm.”

6) The mapping of different domains of Hsp27 to different regions of pK19 is fascinating and promising. Could you please justify why pK19 was chosen, and not pTau40?

We thank the reviewing editor for the positive remarks on our approach using cross-linking mass spectrometry (CL^-^MS) to map the interface of Hsp27 binding to pK19. In the manuscript, we firstly used the BLI assay to measure the binding affinity of Hsp27 to pTau40 and pK19, respectively. The results show that Hsp27 bind to pTau40 and pK19 with a similar affinity, 1.58 VS. 1.98 µM for the equilibrium dissociation constant. This result strongly suggests that pK19 serves as the key region for pTau40 to bind Hsp27. Thus, to probe the key binding region on Hsp27, we used pK19 which contains less amino acids but keeps the key region of pTau40 for binding Hsp27 to simplify the data analysis in CL^-^MS. Accordingly, we revised the description related to this point in the revised Results, section “N-terminal domain (NTD) of Hsp27 mediates the binding of Hsp27 to pTau”, paragraph 2, as following:

“Next, we performed cross-linking mass spectrometry (CL^-^MS) to map the interacting regions of Hsp27 and pK19. pK19 was used since it contains less amino acids but keeps the key region of pTau40 for binding Hsp27 to simplify the data analysis. …”